# Evaluation on Improvement Effect of Different Anti-Stripping Agents on Pavement Performance of Granite–Asphalt Mixture

**DOI:** 10.3390/ma15030915

**Published:** 2022-01-25

**Authors:** Yali Ye, Yan Hao, Chuanyi Zhuang, Shiqi Shu, Fengli Lv

**Affiliations:** 1School of Transportation and Civil Engineering, Shandong Jiaotong University, Jinan 250357, China; 204068@sdjtu.edu.cn (Y.Y.); sdjthaoyan@163.com (Y.H.); lvli7607@163.com (F.L.); 2School of Transportation Engineering, Shandong Jianzhu University, Jinan 250101, China; shiqishu98@gmail.com

**Keywords:** granite–asphalt mixture, adhesion evaluation, Hamburg wheel-tracking test, 1/3 scale accelerated loading test, high-temperature and water stability, anti-stripping measures

## Abstract

There are abundant granite reserves in China, but the adhesion between granite and asphalt is poor, and there are problems such as insufficient water stability, which seriously restrict the application and promotion of granite in asphalt pavement. In order to improve the adhesion between granite and asphalt, as well as the water stability of asphalt mixture, amines and polymers were selected as anti-stripping agents. First, silane coupling agent modified asphalt (SCAMA), rock asphalt modified asphalt (RMA), SBS modified asphalt (SBS), and double rock composite modified asphalt (SCA&RMA) were produced; the modification effect of different anti-stripping modified asphalts was evaluated. Then, the adhesion of different types of asphalts and granite aggregates before and after aging was evaluated by time-delayed water immersion method. Finally, AC-10 and AC-16 granite–asphalt mixtures were designed, through indoor performance test and 1/3 scale accelerated loading test, evaluating the improvement effect of granite–asphalt mixture on pavement performance. The results show that the asphalt modified by amine or organic polymers anti-stripping agent could significantly improve the adhesion between granite and asphalt. The Hamburg wheel-tracking test failed to fully reflect the whole process of high-temperature rutting failure. When evaluating the high-temperature performance and water stability of asphalt mixtures, it is recommended that the evaluation method should cover the whole failure stage of asphalt mixtures; considering the coupling effect of water and high temperature, the order of water stability of granite–asphalt mixture is proposed as follows: SCA&RMA > RMA > SBS > SCAMA > 70-A, and SCA&RMA has the best modification effect.

## 1. Introduction

China is rich in granite reserves; however, the adhesion between granite and asphalt is poor, and there are some problems such as insufficient water stability, which seriously restricts the application and popularization of granite in asphalt pavement. Granite is a kind of deeply igneous rock, and the content of SiO_2_ is about 60–85%. The granite belongs to acidic aggregate, so its adhesion to asphalt is very poor. It is difficult to ensure its water stability and durability when paving asphalt pavement as mix aggregate. The granite–asphalt pavement paved by conventional methods has the problems of asphalt film stripping aggregate, particle falling, looseness, and other water damage. Moreover, the durability of its conventional anti-stripping measures is very poor, which greatly shortens the maintenance cycle of asphalt pavement, increases the maintenance cost, and seriously affects the operation quality of highway [1,2]. Therefore, in order to solve this problem, most companies modified the granite–asphalt mixture or structural parameters by optimizing the pavement structure and material design [3,4].

Many previous studies have improved the adhesion between acid aggregate and asphalt. Measures to improve the resistance of asphalt mixtures to water damage and test methods for evaluating the water resistance of asphalt mixtures have made many referential achievements after a long period of research, and many papers and technical specifications have been published in this regard [5]. Hveem (1937) proposed that the key factor affecting resistance to water damage of acid asphalt mixture was the interface adhesion between asphalt and acid aggregate. At the same time, Marshall engineers suggested using the Marshall immersion test to check the water stability of asphalt mixtures, which had been widely used so far, and this method was also adopted in Chinese specifications [6]. The Federal Highway Administration (1982) pointed out that the water resistance of asphalt mixture made with slaked lime powder and liquid anti-stripping agent was better than that of asphalt mixture made with single anti-stripping agent [7]. The U.S. Strategic Highway Research Program (SHRP) (1988) proposed using hydrated lime powder to replace some mineral powder and adding amine anti-stripping agent to asphalt to improve the water resistance of asphalt [8]. The specification of highway asphalt mixture in Japan put forward that anti-stripping agents such as lime, cement, amine, and amide can be used, and anti-stripping agent only decreased the anti-stripping performance of the mixture [9].

The over-exploitation of high-quality road building materials such as basalt and limestone in China has caused price rises and demand for non-renewable sand and gravel resources. Some domestic scientific research institutions and universities have studied acid aggregates, and the research and use of granite in China are increasing. Xiao et al. (2004) showed that the adhesion promoters could dramatically improve the performance of asphalt concrete of both acidic granite stone and neutral basalt stone. The asphalt added with coupling agent had an adhesion grade of 5 to the stone [10]. Chen Shi and Sha Aimin (2008) took granite in the Pearl River Delta of Guangdong Province as the research object. After adding SBS modifier to original asphalt, the adhesion was improved. It was found that the greater the acid value of asphalt, the better the adhesion with aggregate [11]. Zhang Aiqin (2008) proposed that when the content of titanate coupling agent was 0.45%, the adhesion between granite and asphalt reached grade 5, and the water stability was significantly improved [12,13]. Fan Liang (2009) found that adding an appropriate amount of rock asphalt could effectively improve the high-temperature performance and water damage resistance of an asphalt mixture and that the dynamic modulus of an asphalt mixture would also be improved accordingly [14]. Zhang Wentao (2016) used a variety of stripping agent combinations (Xi’an Huaze PA-A, Shanghai TJ-006 anti-stripping agent, KB-203 dispersant) to modify asphalt. Through experimental analysis, it was proposed that anti-stripping agent had a significant impact on improving the adhesion of asphalt and granite [15]. Adhesion between granite and asphalt increased after SBS-modified asphalt was adopted by Li Mingting (2017) [16]. Peng Chao (2017) studied the adhesion energy and peel energy of silane coupling agent and the interaction between asphalt and aggregate by Fourier transform infrared (FTIR) and scanning electronic microscopy (SEM), indicating that silane coupling agent has a superior effect in improving the adhesion of asphalt aggregate [17]. Han Sen (2019) established a bond failure model of asphalt aggregate based on the surface free energy method to evaluate the improvement effect of hydrated lime on asphalt bond performance. Through a routine performance test and RTFO and PAV aging test, the results showed that hydrated lime could reduce the polarity difference between asphalt and aggregate, increase surface energy, and improve asphalt adhesion [18]. Wang Fuqiang evaluated the influence of silane binder and hydrated lime on improving the water stability of wetland granite–asphalt mixture and proposed that the composite mixing effect of hydrated lime and silane coupling agent was the best [19]. Some researchers have proposed a variety of spalling agent combinations, the compound mixing effect of hydrated lime and silane coupling agent, plant ash byproduct, or recycled paper mill sludge as adhesion promoter to use granite aggregate to improve asphalt mixture performance [20]. Yin Yanping analyzed the effect of aggregate composition on adhesion between aggregate and asphalt [21,22]. XPS spectroscopy was used to characterize the chemical bonds on the surface of the sample. Various macro and micro test methods were used to study the effect of KH-792 silane coupling agent on the surface properties of acidic aggregates and the performance of asphalt mixtures [23]. Lv Songtao (2020) modified the surface of the aggregate with silane coupling agent. Based on the microscopic analysis, Si–O–C and Si–O–Si covalent bonds and hydrogen bonds were generated between the silane coupling agent and the aggregate. It was observed that a polysiloxane coupling layer film was formed on the surface of the aggregate, which improved the surface properties of the aggregate [24]. A molecular dynamic simulation was employed to characterize the asphalt–aggregate interface from the molecular scale by Ding [25]. J. Valentin (2021) proposed that additives to improve the performance of asphalt pavement could not be evaluated based on the results of a single test. There was a correlation between asphalt and aggregate, the fracture mechanics of asphalt concrete, and the sensitivity of mechanical property degradation. The results showed that the silane coupling agent had a good adhesion effect [26].

Previous studies have shown that silane coupling agent has an obvious effect in improving acid aggregate adhesion and water stability. However, at present, the experimental research on silane coupling agent mainly focuses on the rheological properties or microstructure of asphalt mortar, and there is no systematic and comprehensive evaluation on the road performance index of asphalt mixture and different environmental conditions. And the laboratory test is affected by environmental conditions, specimen size, load frequency, and so on, which is quite different from the field pavement. It is difficult to quantitatively evaluate the water damage resistance of silane coupling agent-modified asphalt mixture. Therefore, this paper takes an MTS-810 hydraulic servo testing machine and 1/3 scale pavement accelerated loading test system as the basic experimental platform and forms AC-10 and AC-16 double-layer asphalt mixture composite specimens through the preparation of silane coupling agent, rock asphalt, and other composite anti-stripping agent measures. Through immersion Marshall test and water damage sensitivity analysis test and freeze–thaw splitting test, the degree of contribution and water stability of different anti-stripping agent schemes are analyzed, and the water damage resistance of SCA&RMA mixture is evaluated. It provides a reference for the pavement performance of granite–asphalt mixture.

The research program and process of this paper is shown in Figure 1.

## 2. Materials and Methods

### 2.1. Materials

Qilu70-A asphalt was chosen as base asphalt, and Binhua SBS, Qingchuan rock asphalt, Silane coupling agent (KH-550), and composite agents were taken as anti-stripping measures. The coarse aggregate was adopted granite; its particle size was 10–16 mm, 5–10 mm, and 3–5 mm, respectively. The fine aggregate was 0–3 mm. The mineral filler was limestone powder and lime powder. The silane coupling KH-550 was produced in Zibo, Shandong Province, which is an alkaline, brown–yellow transparent liquid, molecular weight of 220, density of 0.942 g/cm^3^, boiling point of 215 °C, soluble in organic solvents. The technical indexes of materials are shown in Table 1, Table 2, Table 3, Table 4, Table 5, Table 6 and Table 7.

### 2.2. Modification Scheme and Preparation of Modified Asphalt Binders

The original asphalt-related tests were carried out on 70-A asphalt, 0.3% dose of SCAMA, and 5% dose of RMA, SCA&RMA, and SBS modified asphalt binder.

#### 2.2.1. RMA

First of all, Qingchuan rock asphalt was added to the 70-A base asphalt, pre-heated at 145 °C. A stirring process for about 40 min was followed at 150–165 °C. Then the rock asphalt was stirred continuously for 20 min, developed in the oven for 1 h, and followed by another 15 min stirring. The whole process was carried out at 165 °C [27].

#### 2.2.2. SCAMA

The 70-A base asphalt was heated to 145 °C, and the stirring temperature was set to 150 °C. Silane coupling agent was added while stirring for about 50 min until the temperature rose to 160 °C. Subsequently, silane coupling agent was stirred continuously for 30 min at 160 °C and developed in the oven at 80 °C for 2 h.

#### 2.2.3. SCA&RMA

First, 70-A base asphalt was heated to 145 °C. The stirring temperature was first set to 150 °C and increased to 160 °C with the addition of the silane coupling agent. The silane coupling agent was added after 50 min of adding silane coupling agent. Silane coupling agent was added while stirring. Qingchuan rock asphalt (40 min of adding rock asphalt) was added after 160 °C of adding silane coupling agent. The rock asphalt was added after 165 °C and stirred for 30 min. The oven at 165 °C was set for 1 h.

### 2.3. Tests for Asphalt Binder with Different Modification Schemes

The conventional parameters for 70-A, SCAMA, RMA, SCA&RMA, and SBS were measured based on the Standard Test Methods of Bitumen and Bituminous Mixtures for Highway Engineering of JTG E20-2011 [28].

Dynamic shear rheological test [29] and flexural creep stiffness test [30] were carried out in accordance with AASHTO T315-12 and AASHTO TP 125-16. DSR strain control mode was adopted, the target strain value was controlled to 12%, sinusoidal load was applied, the test temperature range was 64–82 °C, one level every 6 °C, and the test frequency was 10 rad/s. The SHRP specification stipulates that when it is 10 °C higher than the minimum pavement design temperature, the creep stiffness s ≤ 300 kPa and creep rate M ≥ 0.3 for 60 s. The experimental temperature adopted by BBR was 18~0 °C and a gradient of 6 °C.

The anti-stripping performance was compared and analyzed by the delayed water immersion method. According to Standard Test Methods of Bitumen and Bituminous Mixtures for Highway Engineering of JTG E20-2011, the water immersion time was to be 0.5 h, 1 h, and 1.5 h.

Short-term aging was simulated by a Rotating Thin Film Oven Test (James Cox & Sons, Colfax, CA, USA) [31]. Simulation of the long-term aging process was carried out by Pressure Aging Vessel (Prentex Alloy Fabricators, Dallas, TX, USA) [32]. The standard aging process of RTFOT was 163 °C and 75 min, and the standard aging process of PAV was 100 °C, 2.1 MPa, 20 h. A conventional asphalt test and dynamic shear rheology (DSR) test were performed on asphalt samples before and after aging [33]. PAV and RTFOT were performed in accordance with the standard methods of AASHTO R28-12 and AASHTO T240-13.

### 2.4. Pavement Performance Tests for Granite–Asphalt Mixture

In this research, the upper layer was AC-10 mixture and the lower layer was AC-16 mixture. The final determination of the optimal aggregate gradation of AC-10 was: (gravel 5–10 mm): (gravel 3–5 mm): (gravel 0–3 mm): mineral powder = 44%: 12%: 40%: 4%. The optimal aggregate grading of AC-16 was: (gravel 10–16 mm): (gravel 5–10 mm): (gravel 3–5 mm): (gravel 0–3 mm): mineral powder = 22%: 40%: 7%: 27%: 4%.

Based on the volume index and mechanical index test of Marshall specimens formed by granite–asphalt mixture AC-10 and AC-16, the AC-10 asphalt aggregate ratio under base asphalt was determined to be 5.1%. In the trial mixing process, it was found that after adding Qingchuan rock asphalt, the color of granite–asphalt mixture was uniform, and the fluidity was significantly reduced. Therefore, the influence of too little asphalt on the research results should be avoided, so the optimal asphalt aggregate ratio was appropriately adjusted to 4.9%.

#### 2.4.1. Anti-Stripping Measures and Schemes

Based on AC-10 asphalt mixture and AC-16 asphalt mixture, the road performance improvement effect of the mentioned anti-stripping measures was evaluated. The scheme of anti-stripping measures is shown in Table 8.

#### 2.4.2. Water Stability Test

According to Standard Test Methods of Bitumen and Bituminous Mixtures for Highway Engineering of JTG E20-2011, the immersion Marshall tests and freeze–thaw splitting test of AC-10 and AC-16 were adopted [28]. The size of the test specimen was the same as that of the standard Marshall test specimen.

The water damage sensitivity tester was used for the water immersion sensitivity test of the granite–asphalt mixture with different anti-stripping measures [34]. The test process was carried out according to AASHTO TP 140-20 standard, and the test parameters were 3500 cycles, 140 °f (60 °C), 40 pounds. Because of the different types of asphalt selected, the heating temperature of asphalt mixture cannot be generalized. Various heating temperatures of asphalt mixtures are shown in Table 9.

#### 2.4.3. Low Temperature Performance of Granite–Asphalt Mixture

According to Standard Test Methods of Bitumen and Bituminous Mixtures for Highway Engineering of JTG E20-2011, a low-temperature bending test was carried out. The temperature was −10 °C, and the loading rate was 50 mm/min, then three-point bending fatigue under split point loading was adopted.

A prismatic specimen with specimen size of 30 mm × 35 mm × 250 mm was adopted, and MTS-810 equipment was used.

#### 2.4.4. Hamburg Wheel-Track Test

According to the AASHTO T324-04 standard, a Hamburg wheel-track test was used to comprehensively evaluate the high-temperature and water resistance of the granite–asphalt mixture [35]. The size of the specimen was Φ150 mm × 60 ± 2 mm, and the water temperature was 50 °C. The loading times were 20,000, 52 ± 2 times/min, and the maximum specimen deformation was controlled at 20 mm, with fixed load 685 N and wheel pressure 0.73 MPa [36].

The double-layer composite specimen of AC-10 upper layer and AC-16 lower layer was adopted in this paper. Since the lower layer AC-16 asphalt was only 70-A and the filler was all mineral powder/lime powder (4:3), the difference between the double-layer mixture specimen schemes lies in the asphalt types and filler types of upper layer AC-10. The test schemes are shown in Table 10.

#### 2.4.5. Mechanical Property

According to Standard Test Methods of Bitumen and Bituminous Mixtures for Highway Engineering of JTG E20-2011, the mechanical properties of asphalt mixture were measured.

The repeated loading times are the corresponding loading times when the permanent deformation reaches 50,000 micro-strain. However, since the SPT simple performance testing machine was used this time, the repeated loading times of the testing machine could only reach 20,000 times; that is, the conditions for the termination of the test were: ① the repeated loading times do not reach 20,000 times, but the structure reaches 5000 micro-strain; ② the structure has not reached 5000 micro-strain, and the number of repeated loading has reached 20,000 times. Therefore, the test was based on the total plastic deformation of asphalt mixture specimen when the loading times were 20,000 times or the loading times when the maximum permanent deformation was 5000 micro-strain [28].

### 2.5. Accelerated Loading Test

An accelerated loading test system can carry out a rapid loading test on pavement or materials in a short time, simulate the damage effects of traffic load on pavement structure for more than ten years, and provide a basis for pavement design, construction, and maintenance. It is the most effective means for evaluating pavement structure design and pavement material performance in the world; the accelerated loading test system is shown in Figure 2.

The accelerated loading test was conducted according to the loading mode of the rutting test in Standard Test Methods of Bitumen and Bituminous Mixtures for Highway Engineering of JTG E20-2011. The size of the specimen was the same as that of the rutting specimen.

The 1/3 scale accelerated loading test system (Shandong Jiaotong University, Jinan, China) is a kind of scale-accelerated loading test facility; it not only can control the temperature and humidity in the enclosure but also can simulate the rainfall process, test the noise, and simulate the sunshine aging process, but also operate all-weather. Four 30 cm × 30 cm × 5 cm specimens can be loaded simultaneously in the accelerated loading test tank. In this paper, the wheel load was 40 kN, and the tire pressure was 0.8 MPa, which is equivalent to the uniaxial load of a double wheel set of 200 kN. The outer diameter of the tire was 380 mm, and the rolling speed was 13 km/h; the rolling length was 600 mm, and 4500 single wheel loads were applied to the specimen every hour.

In order to truly simulate the actual stress state of pavement, accelerated loading failure testing was carried out on the composite asphalt mixture structural specimens of upper AC-10 (3 cm) and lower AC-16 (5 cm). The temperature of air bath was 60 °C and that of the water bath was 50 °C, and the high-temperature performance and water damage resistance sensitivity were evaluated respectively. The high-temperature performance of the granite–asphalt mixture with different anti-stripping measures was compared. The test scheme is shown in Table 10.

#### 2.5.1. Test Specimen

Forming a 300 mm × 300 mm × 80 mm rutting board, the 5 cm thick AC-16 lower layer was paved first, a 3 cm thick aluminum plate was placed for rolling forming, a 3 cm thick aluminum plate was taken out, and then a 3 cm thick AC-10 upper layer was paved on a 5 cm thick AC-16 lower layer after an interval of 5 h, which was rolled and formed in layers, as shown in Figure 3.

#### 2.5.2. Test Procedure

Before testing, the rutting plate was placed in the test groove according to the principle that the traveling direction of the loading wheel is consistent with the rolling direction of the rolling wheel when the rutting plate is formed (as shown in Figure 3). Heat preservation was 5–10 h, and then the loading test was carried out.

After the thermal insulation was completed, the accelerated loading test was started, the speed was set as 13 km/h and the axle load as 40 kN, and the rutting depths were measured every 45,000 times. The rolling was to be terminated when the rutting depth was equal to 20 mm, or the test was to be ended when the rolling times were 72,000 times.

#### 2.5.3. High-Temperature-Coupled Water Stability Evaluation

The 1/3 scale accelerated loading test equipment was adopted, and the specimens were still an 80 mm thick double-layer asphalt mixture. The test groove was filled with 50 °C water, and the liquid level was flush with the top of the test piece. The accelerated loading test was carried out in a water bath environment. When the rutting depth exceeded 20 mm or the rolling times exceeded 72,000 times, the test was to be suspended. The rutting depth at three points (8 cm, 15 cm, and 22 cm) along the traveling direction of the rolling wheel and the rutting plate along the rolling direction were measured every 45,000 times.

## 3. Results and Discussion

### 3.1. Conventional and Rheological Properties of Asphalt Binder

#### 3.1.1. Conventional and Rheological Properties of Original Asphalt Binder

The conventional and rheological properties of the original asphalt binder are shown in Table 11.

It can be seen that the ductility of SCA&RMA was larger than RAM; it can also be observed that the silane coupling agent in SCA&RMA made up for the disadvantage of low-temperature of RMA. The 70-A had the lowest penetration index and the worst temperature sensitivity. The penetration index of SBS was positive, which indicated its temperature sensitivity was the lowest.

At the same temperature, the complex modulus of SCA&RMA was higher than SCAMA and RMA, which showed that the composite superposition of rock asphalt and silane coupling agent produced a positive effect. At the same temperature, the smaller the phase angle was, the better the elasticity was. The deformation resistance was sorted as follows: SBS > SCA&RMA > SCAMA > RMA > 70-A.

At the same temperature, RMA had the largest creep stiffness, which indicated that Qingchuan rock asphalt had increased the brittleness of the original asphalt. Low-temperature crack resistance was sorted as follows: SCAMA > SCA&RMA > RMA, which indicated that the addition of silane coupling agent had been made up for by the poor low-temperature crack resistance of RMA.

#### 3.1.2. Conventional and Rheological Properties of Aged Asphalt Binder

The conventional and rheological properties of aged asphalt binder are shown in Table 12.

It can be seen from Table 12 that short-term aging had little effect on the performance of RMA, SCAMA, and SCA&RMA at high-temperatures. The penetration ratio of SCA&RMA after aging was the largest; it indicates that the aging performance of SCA&RMA was higher than that of single anti-stripping agent modified asphalt.

At the same temperature, the complex modulus of modified asphalt after short-term aging was significantly higher than that of base asphalt, which indicates that the addition of anti-stripping agent improved the deformation resistance of base asphalt.

After long-term aging, the fatigue performance of modified asphalt with anti-stripping agent was significantly higher than that of base asphalt, and the fatigue factor of SBS modified asphalt was the largest. Fatigue resistances were sorted as follows: 70-A < SCAMA < RMA < SCA&RMA < SBS.

### 3.2. Adhesion Evaluation between Granite Aggregate and Asphalt

The delayed water immersion method was used to evaluate the adhesion performance of base asphalt and four anti-stripping modified asphalts. The test results are shown in the Figure 4. It can be seen that the adhesion between 70-A asphalt and granite aggregate was the worst and that the adhesion grade decreased with the extension of water immersion time. The adhesion grade of SBS modified asphalt, RMA, SCAMA, and SCA&RMA were grade 5, which indicates that the asphalt modified by anti-stripping agent had good adhesion with granite.

In order to further evaluate the adhesion between asphalt and granite aggregate after aging, The short-term and long-term aging tests of 70-A asphalt and four modified asphalts were carried out respectively; the short-term aging test was carried out with the Rolling Thin Film Oven Test (RTFOT), and the long-term aging test was performed with Pressure Aging Vessel (PAV). The delayed water immersion test was carried out on the aged asphalt, and the delay time adopted 3 min, 15 min and 30 min, respectively. The results of the short-term aging test and long-term aging test are shown in Figure 5 and Figure 6, respectively.

It can be seen that adhesion grades were sorted as follows: SCA&RMA > RMA, SCAMA > SBS > 70-A. After long-term aging, the adhesion grade between 70-A asphalt and granite was only grade 1 at 30 min, while SCA&RMA, RMA, and SCAMA could reach grade 4. This is because the addition of silane coupling agent or rock asphalt improved the intermolecular force, enhanced the polar bond, and improved the adhesion with aggregate.

### 3.3. Water Stability Evaluation

According to the scheme in Table 10, the water stabilities of AC-10 and AC-16 granite–asphalt mixtures were evaluated through water immersion Marshall test, freeze–thaw splitting test, and water damage sensitivity analysis test.

#### 3.3.1. Water Immersion Marshall Test

The results of immersion Marshall test are shown in Table 13.

It can be seen from Table 13 that the stability, flow value, and residual stability of AC-10 and AC-16 granite–asphalt mixtures followed the same law. The residual stability of each scheme was greater than 85%, which met the requirements of specification. According to the residual stability index, water stability was sorted as follows: scheme 5 > scheme 3 > scheme 2 > scheme 4 > scheme 1.

This is because Qingchuan rock asphalt increased the content of nitrogen in the form of polar functional groups. The addition of silane coupling agent combines organic functional groups with granite aggregate, so as to organize the surface of aggregate and greatly improve the water stability of the granite–asphalt mixture.

#### 3.3.2. Freeze–Thaw Splitting Test

After three freeze–thaw cycles, the residual strength ratio of freeze–thaw split test are shown in Figure 7.

The residual strength ratio of different schemes was greater than 80%, which met the requirements of specification. According to TSR index, water stability was sorted as follows: scheme 5 > scheme 3, scheme 2 > scheme 4 > scheme 1, The results were basically the same as the immersion Marshall test. The residual strength of the freeze–thaw splitting test in scheme 5 was better than TSR. Scheme 1 could not meet the specification requirements after three freeze–thaw cycles, while the other four schemes could meet the specification requirements, indicating that scheme 1 had the worst durability. This is because although the freeze–thaw splitting effect had met the requirements at the beginning, it was very easy to peel after many freeze–thaw cycles, and water damage such as pits and grooves had been formed.

#### 3.3.3. Water Damage Sensitivity Analysis Test

The water damage sensitivity test (MIST) was used to test the sensitivity of asphalt mixtures to water damage of various anti-stripping measures. The results of MIST residual stability are shown in Figure 8.

After 3500 times of scouring, the residual stabilities of AC-10 and AC-16 under different anti-stripping schemes are shown in Figure 6. Taking AC-10 as a representative, compared with the test results of the conventional immersion Marshall test, the residual stability was decreased, and scheme 5 (SCA&RMA) had the least reduction in residual stability.

Figure 8 shows the residual strength ratio of asphalt mixture with different anti-stripping measures after MIST scouring.

From Figure 9 it can be seen that the residual strength ratios of the AC-10 and AC-16 were similar to those of the conventional freeze–thaw splitting test. The difference was that after the freeze–thaw splitting test, the residual strength ratio of the five schemes all met the requirements of specification. After MIST, only scheme 2 (SCAMA) and scheme 5 (SCA&RMA) satisfied the requirements of specification (≥80%). It proved that when evaluating the water stability of asphalt mixtures, it is not only necessary to consider its freeze-thaw effect or water immersion effect, but also necessary to consider the water stability of the asphalt mixture under dynamic water pressure. Only considering the water stability evaluation of freeze–thaw or water immersion effect will not truly reflect the impact of water damage to the asphalt mixture.

### 3.4. Low-Temperature Crack Resistance

The results of low-temperature bending test are shown in Table 14.

From Table 14 it can be seen that the flexural tensile strength was sorted as follows: scheme 3 (SBS) > scheme 5 (SCA&RMA) > scheme 2 (SCAMA) > scheme 1 (70-A) > scheme 4 (RMA). It can be found that scheme 3 (SBS) was the best, while scheme 4 (RMA) was the worst, and only scheme 4 did not meet the specification requirements (>2000 με). The analysis showed that the organic functional group of silane coupling agent molecule was combined with the functional group of inorganic materials such as stone. In addition, its inorganic functional group was different from that of organic materials such as asphalt. Chemical or physical winding would firmly bond the two materials with very different properties in the asphalt mixture so as to establish a “molecular bridge” with special functions between inorganic stone and organic polymer asphalt, which could transfer stress between stone and asphalt, so as to strengthen the adhesion between stone and asphalt, so as to enhance the low-temperature performance of the asphalt mixture.

### 3.5. High-Temperature Rutting Resistance

Figure 10 shows the dynamic stability of asphalt mixture with different anti-stripping measures.

Table 10 shows the influence of five schemes on the dynamic stability of AC-10 and AC-16 and that they all met the requirements of specification (>2800 times/mm). Taking AC-10 as a representative, the impact of five schemes on the performance of high-temperature asphalt mixtures were analyzed. Scheme 1 (70-A) was the worst, and its dynamic stability was only 2971 times/mm; scheme 3 (SBS) provided the best results due to the high softening temperature and low-temperature sensitivity of SBS modified asphalt. The dynamic stability value of scheme 5 (SCA&RMA) was between scheme 4 (RMA) and scheme 2 (SCAMA), which showed that SCA&RMA successfully coated the granite surface and improved the performance of the asphalt mixture at high-temperatures.

### 3.6. High-Temperature and Water Stability of Double-Layer Composite Specimen

As mentioned above, a Hamburg wheel-tracking test was used to comprehensively evaluate the high-temperature stability and water stability of the granite–asphalt mixture; the test results are shown in Figure 11.

It can be drawn from Figure 11 that at the same rolling times, the depths of Hamburg wheel-tracking were sorted as follows: scheme 11 (70-A) > scheme 21 (SCAMA) > scheme 41 (RMA) > scheme 51 (SCA&RMA) > scheme 31 (SBS). In the rutting test, scheme 31 (SBS) had the best rutting resistance; while the Hamburg wheel-tracking test was inferior for scheme 51, it shows that under the coupling action of high-temperature and water, asphalt mixture experiences more severe environmental conditions, and scheme 51 shows better performance than scheme 41 (RMA). This is because the silane coupling agent is a silicone monomer with a silane oxygen group and an organic functional group. The silane oxygen group is reactive to the inorganic matter in granite, while the organic functional group is reactive and mutually inclusive to the organic matter in asphalt. Qingchuan rock asphalt has high nitrogen content and good compatibility with asphalt and silane coupling agent, which enhances the adhesion and peeling resistance with aggregate. Therefore, when the silane coupling agent is at the interface between asphalt and granite, it can react with the organic matter in rock-modified asphalt and the inorganic material of granite to form the binding layer of asphalt organic matrix, silane coupling agent, and granite inorganic matrix, improving the intermolecular force, enhancing the polar bond, and producing chemical cross-linking and polymerization to form macromolecular network structure. Its addition improves the hydrophobicity and adhesion of rock asphalt-modified asphalt polymer.

### 3.7. Mechanical Properties of Granite Double-Layer Composite Specimens

As can be seen from Figure 12:(1)Under the conditions of 20 °C, 35 °C, and 50 °C, respectively, the dynamic modulus of granite–asphalt mixture specimens formed in the same scheme decreased with the decrease in loading frequency, and the five schemes showed the same variation law. The analysis showed that the elasticity of asphalt mixture decreased and that the viscosity increased when the loading frequency decreased gradually. Under the condition of a certain loading frequency, the dynamic modulus decreased with the increase in temperature, which was mainly due to the gradual decrease in asphalt stiffness modulus as binder with the increase in mixture temperature. Under the action of stress, the resilience decreased, which was reflected in the decrease in dynamic modulus.(2)As the temperature increased or the load frequency decreased, the phase angles all increased first, and then decreased after the peak. When exposed to high-temperature and low frequency, the viscosity of the asphalt binder decreased, and the effect on the performance of the asphalt mix was also reduced. At this time, the mineral aggregate skeleton had become the main factor affecting the asphalt mixture, and the mineral aggregate was an elastic material and its phase angle was zero, so the phase angle of the asphalt mixture would decrease.(3)As the loading frequency changed from high to low, the slope of the main curve first increased and then decreased, indicating that the dynamic modulus of the asphalt mixture did not change much with loading frequency under the conditions of extremely high frequency and extremely low frequency. In the intermediate frequency range (the frequency range expanded with the increasing temperature), the loading frequency had a great influence on the dynamic modulus of the mixture.

### 3.8. Performance of Double-Layer Asphalt Mixture Based on Accelerated Loading Test

#### 3.8.1. High-Temperature Stability of Double-Layer Composite Specimens

It can be seen from Figure 13, the test results of accelerated loading rutting depth had the same consistency with the conventional rutting test results. The modified asphalt mixture significantly reduced the rutting depth, except for scheme 21.

The comparison shows that when the number of load tests was 9000, the rutting depth of scheme 31, scheme 41, and scheme 51 was reduced by 29%, 57%, and 55%, respectively. It can be seen that scheme 41 and scheme 51 played an important role in improving the rutting resistance of the asphalt mixture. The rule of changing the rutting depth was basically the same as for the conventional rutting test. The growth of rutting depth basically increased rapidly at first, and then tended to be stable, and increased rapidly again with the increase in rolling times. That is, the permanent deformation change law of the asphalt layer was divided into three phases. The first phase was the rapid growth phase; in this phase, the asphalt mixture was quickly compacted under the action of the upper axle load, and the deformation increased rapidly. The second phase was the steady development stage, in which the asphalt mixture had a certain compactness, the deformation rate changed little, and the deformation developed steadily. The third phase was the accelerated failure stage, in which the deformation rate increased rapidly, and the mixture had large-area shear flow, but the volume did not change.

#### 3.8.2. Performance of Composite Specimens under High-Temperature and Water

High-temperature and water stabilities of double-layer composite specimens based on an accelerated loading test were carried out; the results are shown in Figure 14.

From Figure 14, it can be seen that under repeated wheel load, the initial rutting depth of the asphalt mixture increased rapidly and then stabilized, and that the rutting depth increased rapidly at the beginning of the load, and the deformation speed gradually decreased. Then it entered the stable stage in which the rut depth developed slowly with the increase in wheel load, the deformation rate remained basically unchanged, and then it entered the rapid failure stage. Comparing the rutting depth curves of granite–asphalt mixture under the coupling effects of water resistance and high-temperature in the five schemes, the asphalt structure layer had experienced three stages: initial compaction, medium-term creep development, and basically late instability failure. However, the granite–asphalt mixture with different anti-spalling measures showed great differences in the medium-term creep development stage, except for scheme 51; the other four schemes completed the medium-term creep development stage under the action of 4500–9000 wheel loads and entered the late instability failure stage.

In the 50 °C water bath environment, under conditions of 200 kN axle load (40 kN wheel load), the rutting depth and development speed of granite–asphalt mixture scheme 51 were the smallest, and the peeling breaking point was about 40,500 wheel loads, which was much higher than the 13,500 wheel loads of scheme 21 and 22,500 wheel loads of scheme 31. The higher the peeling point, the less likely it was to produce rutting. The earlier the spalling break point appeared, the worse the anti-spalling ability of the mixture.

Compared with the Hamburg wheel-tracking test, the loading times of this test increased by about 20,000 times, as shown in Figure 14. In the Hamburg wheel-tracking test, when the rolling times were 15,000–20,000 times, the growth rate of the rutting curve increased, although the water bath-accelerated loading test showed basically the same change law. However, after the rutting depth exceeded 20,000 times, the rutting depth of scheme 51 increased slowly with the number of wheel load actions, and the curve slope was small, showing good water stability. Scheme 21 had the fastest creep rate and directly entered the later accelerated failure stage after the initial compaction stage, and the rut depth was the deepest. The rutting depth of scheme 11 basically increased linearly, and the water stability of other schemes was good. An accelerated loading test could more accurately characterize the high-temperature and anti-rutting performance and the performance decayed law of asphalt mixture or pavement.

### 3.9. Discussion

Based on the evaluation of adhesion between asphalt and granite aggregate and the study of pavement performance such as water stability of asphalt mixture, the improvement effects of four anti-stripping agents on the performance of asphalt binder and granite–asphalt mixture were evaluated.

Through the asphalt DSR, BBR, and adhesion test with granite aggregate, it was found that the complex modulus and rutting factor of SCA&RMA increase, the phase angle and creep stiffness decrease, and the composite superposition of rock asphalt and coupling agent has a positive effect, which not only makes up for the defects of poor durability and easy decomposition of silane coupling agent but also makes up for the shortcomings of poor low-temperature crack resistance of Qingchuan rock asphalt. The delayed water immersion method was used to verify that SCA&RMA after RTFOT and PAV had the smallest peeling area and the best adhesion with granite aggregate.

Based on the laboratory pavement performance test, the silane coupling agent could improve water stability and low-temperature crack resistance, but its thermal stability and durability were lower than other anti-stripping measures. Qingchuan rock asphalt showed excellent high-temperature rutting resistance and aging resistance but insufficient low-temperature crack resistance. The effects of SCA&RMA on improving granite–asphalt mixture was optimal in general.

Based on a 1/3 scale accelerated loading test machine, the rutting specimens of double-layer asphalt mixture were designed to evaluate the structural performance of granite–asphalt mixture under the coupling effect of load and environment. Under the combined influence of high-temperature, water, and heavy axle load, the SCA&RMA mixture had the best high-temperature stability and water stability, and it had experienced three processes of initial compaction, medium creep, and late failure. The four anti-stripping measures of RMA, SBS, 70-A, and SCA entered the failure stage only after a small amount of wheel load.

## 4. Conclusions

To improve the adhesion between granite acid aggregate and asphalt, as well as improve the water stability and durability of granite–asphalt mixture, amine and organic polymer anti-stripping agent were selected in this research, and they were made into four kinds of modified asphalt. Based on the laboratory test research and mechanism analysis of asphalt binder and granite–asphalt mixture, the following conclusions were drawn:

Asphalt modified by amine or organic polymer anti-stripping agent or composite can significantly improve the adhesion between granite aggregate and asphalt; the adhesion grades of granite aggregate to different types of asphalt were sorted as follows: SCA&RMA > RMA, SCAMA > SBS > 70-A.

Short-term aging had little effect on the performance of RMA, SCAMA, and SCA&RMA at high temperatures. The complex modulus of modified asphalt after short-term aging was significantly higher than that of base asphalt. Modified asphalt can improve the fatigue performance of asphalt greatly. Fatigue resistance was sorted as follows: 70-A < SCAMA < RMA < SCA&RMA < SBS.

The water sensitivity analysis test could accurately evaluate the water stability of the asphalt mixture, and the water stability of the granite–asphalt mixture was sorted as follows: SCA&RMA > SCAMA > SBS > RMA > 70-A.

Compared with the Hamburg wheel-tracking test, the accelerated loading test tire was closer to the actual vehicle, and the loading times were user-defined, which could be used to completely obtain the whole failure process of the asphalt mixture. The asphalt mixture with slow rutting development in the initial stage of loading entered the failure stage, especially when the loading times exceeded 20,000 times, which affected the evaluation of asphalt mixture road performance. Therefore, the pavement performance of the granite–asphalt mixture should be comprehensively evaluated by improved test methods or test means.

The KH-550 of SCA and RMA selected in this research only introduced color, apparent state, and physical properties and only preliminarily analyzed the physical and chemical mechanism of Qingchuan rock asphalt, coupling agent, and granite aggregate. There was no in-depth correlation analysis between the chemical composition and the pavement performance of the asphalt mixture. The performance of the granite–asphalt mixture affected by high-temperature and water environment was preliminarily assessed by using a 1/3 scale accelerated loading test machine, which was not deep enough. Therefore, it is necessary to carry out more in-depth research on the improvement in adhesion between granite and asphalt and the improvement in the pavement performance of the asphalt mixture on multiple scales.

## Figures and Tables

**Figure 1 materials-15-00915-f001:**
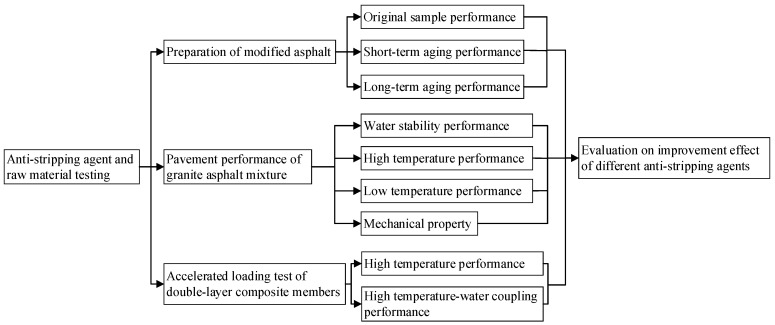
Research program and process.

**Figure 2 materials-15-00915-f002:**
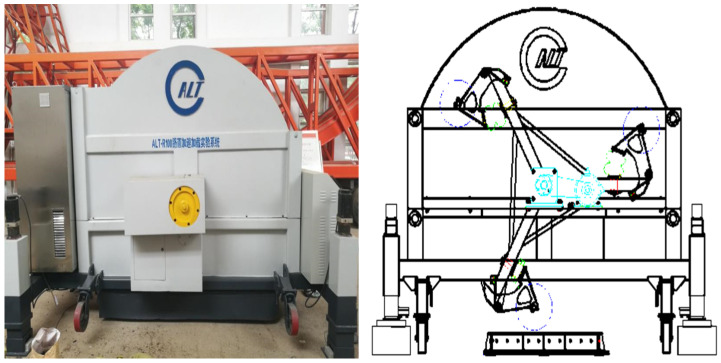
The 1/3 scale accelerated loading test system.

**Figure 3 materials-15-00915-f003:**
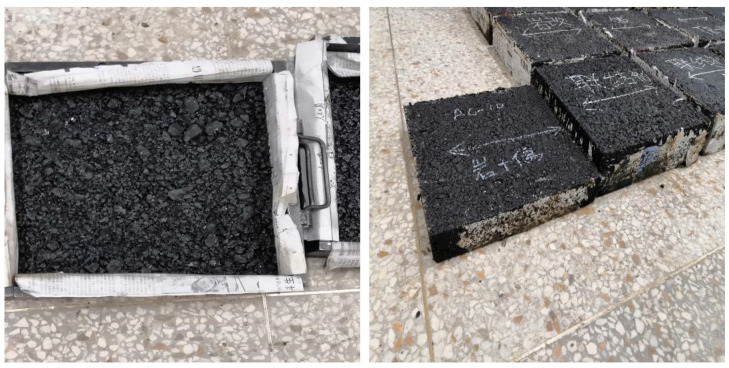
Composite rutting specimen with double-layer asphalt mixture.

**Figure 4 materials-15-00915-f004:**
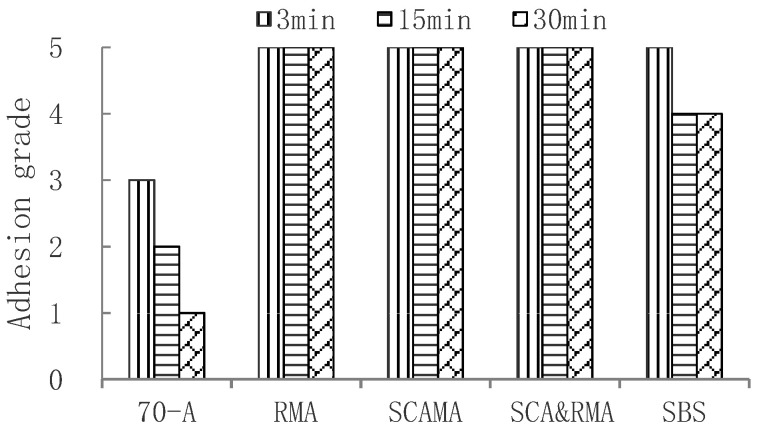
Adhesion grade of original asphalt and granite aggregate.

**Figure 5 materials-15-00915-f005:**
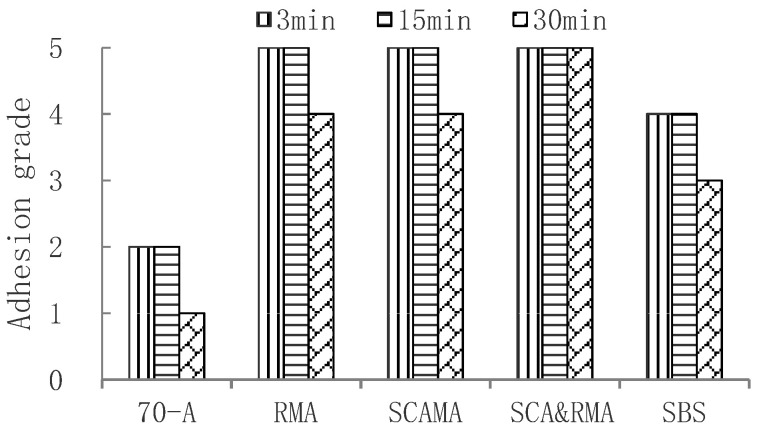
Adhesion grade of granite aggregate and short-term aged asphalt.

**Figure 6 materials-15-00915-f006:**
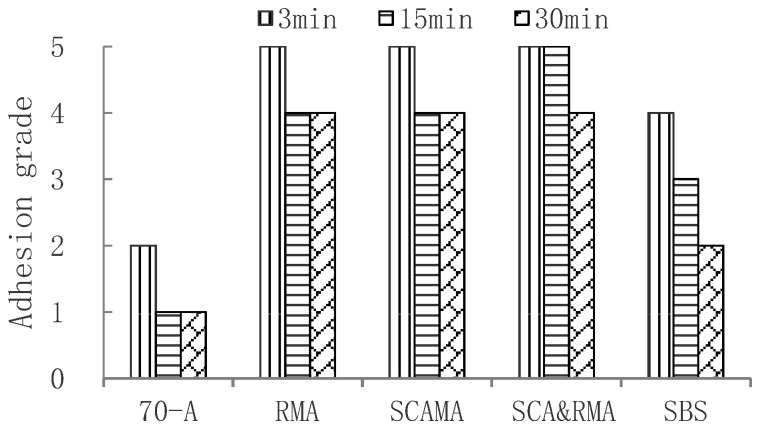
Adhesion grade of granite aggregate and long-term aged asphalt.

**Figure 7 materials-15-00915-f007:**
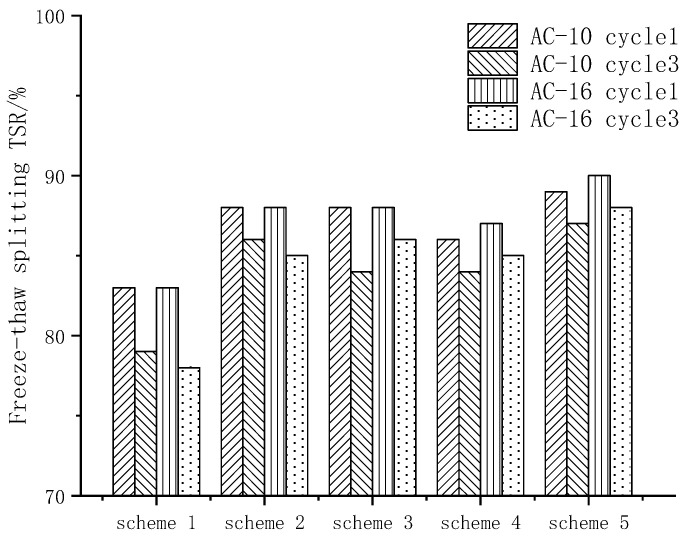
Residual strength ratio of freeze–thaw splitting test.

**Figure 8 materials-15-00915-f008:**
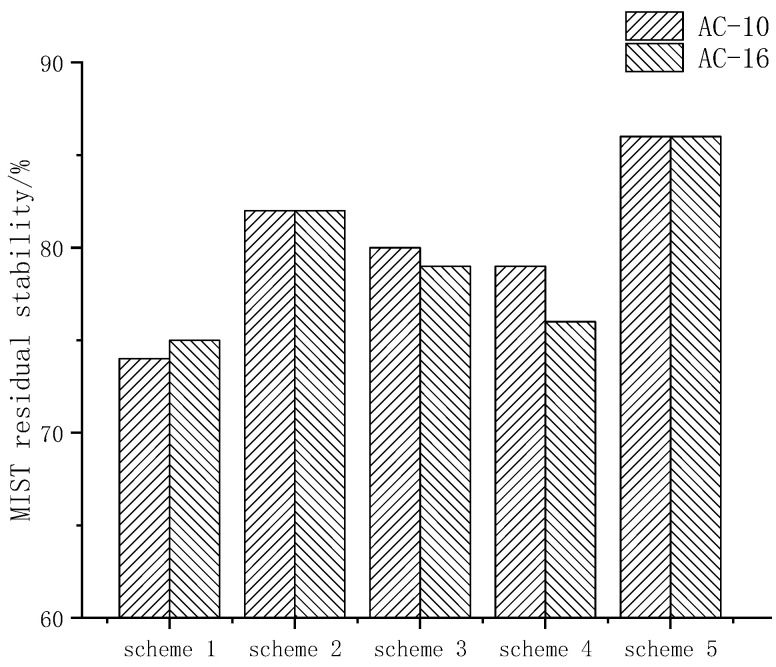
Residual stability by MIST.

**Figure 9 materials-15-00915-f009:**
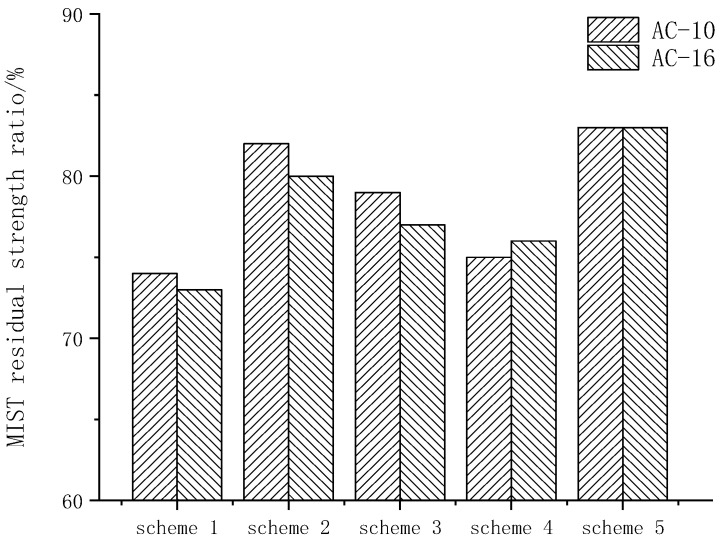
Residual strength ratio by MIST.

**Figure 10 materials-15-00915-f010:**
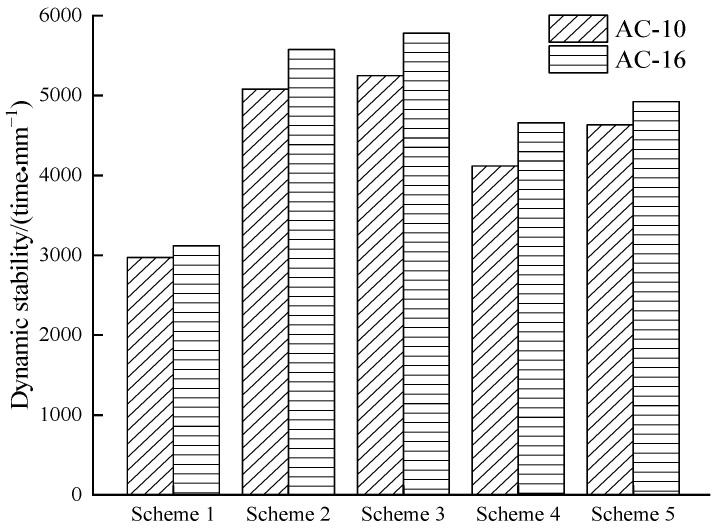
Dynamic stability of AC-10 and AC-16.

**Figure 11 materials-15-00915-f011:**
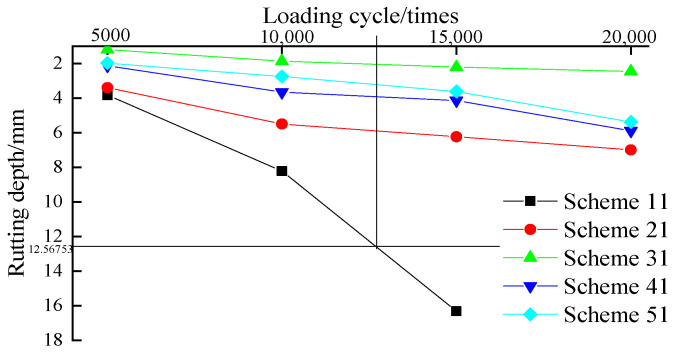
Hamburg curve of rutting depth.

**Figure 12 materials-15-00915-f012:**
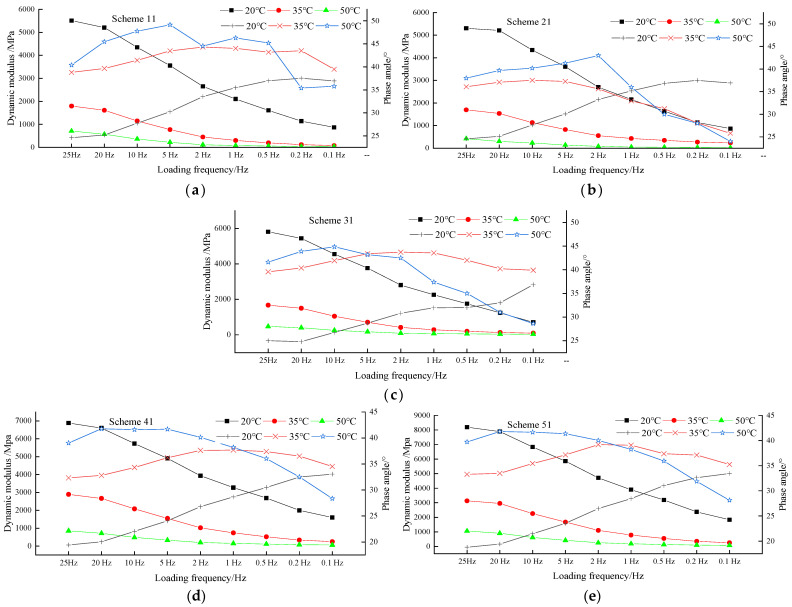
Dynamic modulus and phase angle curves of mixes: (**a**) Scheme 11, (**b**) Scheme 21, (**c**) Scheme 31, (**d**) Scheme 41, (**e**) Scheme 51.

**Figure 13 materials-15-00915-f013:**
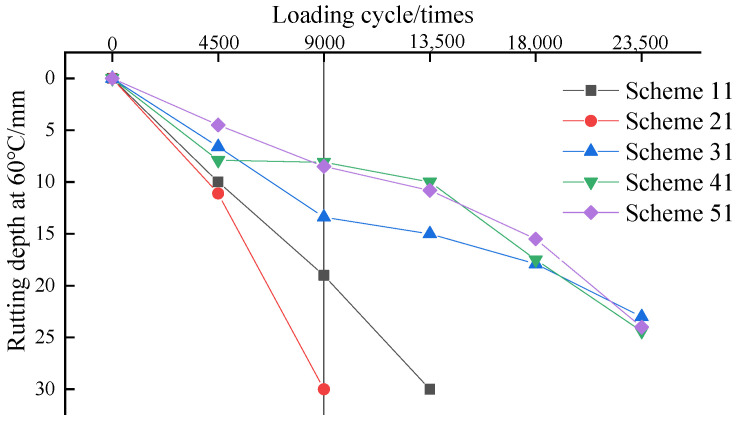
Rutting depth at 60 °C of double-layer asphalt mixture composite specimen.

**Figure 14 materials-15-00915-f014:**
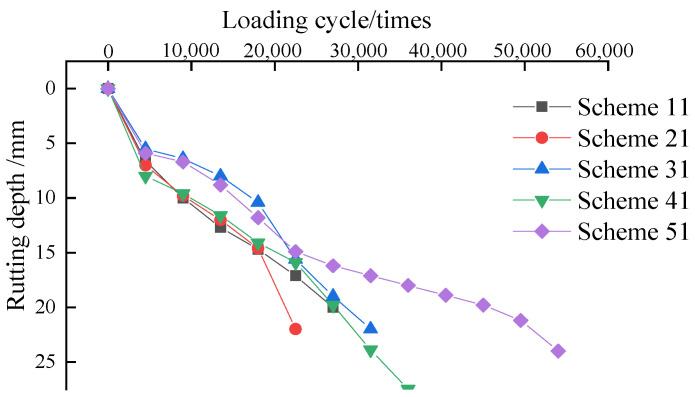
Rutting depth under high-temperature and water.

**Table 1 materials-15-00915-t001:** Technical index of base asphalt and SBS modified asphalt.

Asphalt	Original				RTFOT		
	Penetration Degree (25 °C, 100 g, 5 s)/0.1 mm	PI	Softening Point/°C	Ductility (5 cm/min)/cm	Penetration Ratio/%	Ductility (5 cm/min)/cm	Mass Loss/%
70-A	72.0	−0.81	47.5	38 ①	70	17 ①	−0.10
SBS	55.7	0.15	69.0	25.4 ②	73	18 ②	−0.12

Notes: ① 10 °C; ② 5 °C.

**Table 2 materials-15-00915-t002:** Physical properties of coarse aggregate.

Index		Test Results	Technical Requirement
Aggregate crushing value/%		18.8	≯26
Apparent density	3–5	2.712	≮2.60
	5–10	2.642	
	10–15	2.621	
	10–20	2.630	
Water absorption/%		1.8	≯2.0
Adhesion level		2	≮4 grade
Needle flake particle content/%		9	≯15
Water absorption of fine aggregate/%		0.4	≯1
Soft stone content/%		1.2	≯3
Los Angeles abrasion value/%		19	≯28

**Table 3 materials-15-00915-t003:** Physical properties of fine aggregate.

Index	Test Results	Technical Requirement
Apparent density	2.658	≮2.50
Sediment percentage/%	1	≯3
Sand equivalent/%	87	≮60
Methylene blue value/g/kg	15	≯25

**Table 4 materials-15-00915-t004:** Physical properties of limestone powder.

Index	Apparent Density	Water Content/%	Sieve Size		
			<0.6 mm	<0.15 mm	<0.075 mm
Test results	2.632	0.68	100	95.8	86.4
Technical requirement	≥2.5	≤1.0	100	90–100	75–100

**Table 5 materials-15-00915-t005:** Main technical indexes of Qingchuan rock asphalt.

Technical Indicators	Test Results	Test Method
Appearance	Brown Powder	Visual inspection
Ash content	9.6	JTG E20 T0614
Density/25 °C	1.201	JTG E20 T0603
Water content/%	1.03	JTG E20 T0612

**Table 6 materials-15-00915-t006:** Technical index of binder.

Appearance	Odor	Melting Point	Density
White solid powder	Light acidity	120–140 °C	0.92 g/cm^3^

**Table 7 materials-15-00915-t007:** Technical index of lime powder.

Index	Technical Requirement	Test Results
Apparent density	≥2.5	2.612
Particle size range	<0.6 mm/% 100–100	100.0
	<0.15 mm/% 90–100	97.6
	<0.075 mm/% 85–100	86.7
Moisture content	≤1%	0.4%
Hydrophilic coefficient	≤1	0.4
Plasticity index	≤4	2.8

**Table 8 materials-15-00915-t008:** Scheme of anti-stripping measures for asphalt mixture.

Scheme	Asphalt	Filler
1	70-A	Mineral powder/lime powder = 4:3
2	SCAMA	Mineral powder/lime powder = 4:3
3	SBS	Mineral powder/lime powder = 4:3
4	RMA	Mineral powder
5	SCA&RMA	Mineral powder

**Table 9 materials-15-00915-t009:** Various heating temperatures of asphalt mixture.

Scheme	Asphalt Temperature/°C	Aggregate Temperature/°C	Trial Temperature/°C	Mineral Powder Temperature/°C	Mixing Pot Temperature/°C	Mixing Temperature/°C	Molding Temperature/°C
1	155	170	120	120	165	155	150
2	165	180	120	120	175	165	155
3	170	185	120	120	180	170	160
4	165	180	120	120	175	165	155
5	165	180	120	120	175	165	155

**Table 10 materials-15-00915-t010:** Scheme of double-layer asphalt mixture composite specimen.

Scheme	3 cm AC-10 Surface Layer		5 cm AC-16 Lower Layer	
	Asphalt	Filler	Asphalt	Filler
11	70-A	Mineral powder/lime powder = 4:3	70-A	Mineral powder/lime powder = 4:3
21	SCAMA	Mineral powder/lime powder = 4:3	70-A	Mineral powder/lime powder = 4:3
31	SBS	Mineral powder/lime powder = 4:3	70-A	Mineral powder/lime powder = 4:3
41	RMA	Mineral powder	70-A	Mineral powder/lime powder = 4:3
51	SCA&RMA	Mineral powder	70-A	Mineral powder/lime powder = 4:3

**Table 11 materials-15-00915-t011:** Conventional and rheological properties of original asphalt binder.

Test Projects	Index		70-A	RMA	SCAMA	SCA&RMA	SBS
Conventional properties	25 °C Penetration/0.1 mm		72	52	70	54	55.7
	Softening Point/°C		47.5	52	51	52	69
	Ductility/cm		>100	32	>100	>100	>100
	135 °C Brinell viscosity/Pa·s		0.3	0.91	0.5	0.7	2.13
DSR	G*/kPa	64 °C	1.629	2.489	2.742	3.369	3.62
		70 °C	0.786	1.146	1.263	1.586	2.02
		76 °C	-	0.576	0.626	0.796	1.36
		82 °C	-	-	-	-	0.76
	δ/°	64 °C	87.07	86.36	86.05	85.26	68.62
		70 °C	88.1	87.55	87.21	87	65.01
		76 °C	-	88.37	88.06	87.89	63.37
		82 °C	-	-	-	-	59.82
	G*/sinδ	64 °C	1.631	2.494	2.749	3.381	3.888
	/kPa	70 °C	0.787	1.147	1.264	1.588	2.229
		76 °C	-	0.577	0.626	0.797	1.521
		82 °C	-	-	-	-	0.879
	PG high-temperature grade/°C		64	70	70	70	76
BBR test	S/MPa	−18 °C	356	420	370	382	361.5
		−12 °C	154	226.5	206.5	187	170.5
		−6 °C	79.5	93.7	83.7	86.6	86.2
		0 °C	35.9	43.7	41.7	42	41.4
	m	−18 °C	0.248	0.224	0.244	0.231	0.28
		−12 °C	0.328	0.32	0.33	0.331	0.32
		−6 °C	0.382	0.37	0.38	0.384	0.429
		0 °C	0.438	0.432	0.439	0.436	0.525
	PG low-temperature grade/°C		22	22	22	22	22

**Table 12 materials-15-00915-t012:** Conventional and rheological properties of aged asphalt binder.

	Index		70-A	RMA	SCAMA	SCA&RMA	SBS
Conventional properties after RTFOT	25 °C Penetration/0.1 mm		50.4	33.4	49.5	40	40.7
	Softening Point/°C		40.4	38.1	41	41	52
	Ductility/cm	15 °C	49.9	11.6	50	28	40
DSR after RTFOT	G*/kPa	64 °C	3.14	5.021	5.22	5.62	11.58
		70 °C	1.433	2.316	2.216	2.532	5.653
		76 °C	-	1.116	1.316	1.34	2.347
		82 °C	-	-	-	-	1.467
	δ/º	64 °C	85.32	83.48	83.38	83.32	80.22
		70 °C	86.84	85.42	85.62	85.06	82.35
		76 °C	-	86.87	86.97	86.52	84.23
		82 °C	-	-	-	-	85.78
	G*/sinδ	64 °C	3.15	5.053	5.255	5.658	11.751
	/kPa	70 °C	1.435	2.323	2.222	2.541	5.704
		76 °C	-	1.117	1.318	1.342	2.359
		82 °C	-	-	-	-	1.471
	PG high-temperature grade/°C		64	70	70	70	76
DSR after PAV	G*/kPa	28 °C	-	4932	5231	5514	9328
		25 °C	5118	7176	7567	7960	11780
		22 °C	7283	-	-	-	-
	δ/°	28 °C	-	48.44	47.23	46.36	39.94
		25 °C	47.39	46	45.6	44.92	37.9
		22 °C	43.98	-	-	-	-
	G*sinδ/kPa	28 °C	-	3690	3840	3990	5988
		25 °C	3767	5162	5406	5621	7236
		22 °C	5058	-	-	-	-
	PG medium-temperature grade/°C		25	28	28	28	28

**Table 13 materials-15-00915-t013:** Results of immersion Marshall test.

Scheme Category	AC-10 Results of Immersion Marshall Test				AC-16 Results of Immersion Marshall Test			
	Soaking Time/h	Flow Value/mm	Stability/kN	Residual Stability/%	Soaking Time/h	Flow Value/mm	Stability/kN	Residual Stability/%
Scheme 1	0.5	3.4	18.89	84.8	0.5	3.9	15.66	85.1
	48	4.2	16.03		48	3.6	13.32	
Scheme 2	0.5	3.8	17.17	90.3	0.5	4.4	16.06	89.2
	48	4	15.51		48	4.1	14.32	
Scheme 3	0.5	4.9	22.2	92.2	0.5	4.2	16.06	91.5
	48	4.8	20.47		48	4	14.7	
Scheme 4	0.5	3.6	17.66	87.2	0.5	3.6	17.66	88.9
	48	3.8	15.4		48	3.8	15.7	
Scheme 5	0.5	3.9	20.06	94.1	0.5	3.9	20.06	92.2
	48	4.1	18.88		48	4.1	18.5	

**Table 14 materials-15-00915-t014:** Results of low-temperature bending test.

Mixture Type		Relative Density of Gross Volume	Mid-Span Deflection/mm	Maximum Damage Load/N	Flexural Tensile Strength/MPa	Bending Strain/µε	Bending Stiffness Modulus/MPa
AC-10	Scheme 1	2.286	0.42	798.4	6.3	2214.79	2857
	Scheme 2	2.258	0.56	924.5	7.2	2993.13	2411.73
	Scheme 3	2.339	0.74	1204.6	9.6	3859.41	2486.98
	Scheme 4	2.301	0.36	793.7	6.1	1932.72	3166.52
	Scheme 5	2.37	0.64	1149	9.3	3323.02	2799.17
AC-16	Scheme 1	2.298	0.39	789.5	6.9	2066.41	3322.34
	Scheme 2	2.34	0.59	889.5	7	2415.21	3247.45
	Scheme 3	2.375	0.66	1112.1	7.2	3466.73	2074.52
	Scheme 4	2.383	0.39	924.5	7.1	1896.44	3763.48
	Scheme 5	2.321	0.45	985.2	7.7	2393.87	3211.58

## Data Availability

Data available on request due to restrictions eg privacy or ethical. The data presented in this study are available on request from the corresponding author. The data are not publicly available due to the study data was not links to publicly archived datasets.

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
