# Peer review of "Evaluation on Improvement Effect of Different Anti-Stripping Agents on Pavement Performance of Granite–Asphalt Mixture"

_materials, 2022, doi:10.3390/ma15030915_

Round 1

Reviewer 1 Report

Although I have tried my best, this manuscript is un-readable.  This is not only due to the lack of proper English; a manuscript submitted in this form should be rejected by the editor already, but mainly due to the lack of clarity of presentation. A complete re-writing and re-organization of the manuscript is recommended to qualify for a new re-submission.  Maybe some additional remarks and suggestions/requests which should be followed by the authors when submitting scientific work:

  1. Any experimental work must be described in a way that it can be re-done anywhere. That means that there is a need to provide details about materials (origin, composition etc.) and not just code names like KH -550 for the coupling agent. The mentioning of limestone powder is not enough, details about grain size distribution are needed! How is the rock asphalt characterized, etc! It is unclear what final composition corresponds to AC-190 and to AC-16!
  2. The goal of the investigation should be clearly explained and elaborated followed by a design of the experiments to enable a reader to follow. The authors use, e.g. in table 2 as well as in table 3 the terms scheme 1-6 for totally different things, most confusing!
  3. Following the DoE, a presentation of the results in a clear form should appear. Subsequently, these results may be discussed in view of the DoE, the differences in composition of the samples etc.
  4. Finally in the conclusion part, trends corresponding to components, composition and other influences should be discussed to derive in conclusions.

Reviewer 2 Report

  1. Authors should revise the title to suit the paper content
  2. Please revise the abstract by incorporating essential elements
  3. Please avoid extensively long sentences (several sentences consumed up to 5 lines, the reader need to read repeatedly to understand the meaning)
  4. Please revise the whole paper for the grammatical error and improper sentence structures
  5. Authors should provide the standard specifications referred for each test
  6. Please revise the paper and reference formatting 
  7. A lot of sentences miss space between words
  8. Explanations on the test schemes should be further elaborated
  9. Please check the unit used (inconsistent and different symbols were used, e.g. temperature, MPa instead of Mpa, and a lot more)
  10. Please revise the conclusions section

Reviewer 3 Report

The manuscript entitled " Evaluation of adhesion improvement measures between granite and asphalt and pavement performance of asphalt mixture” presented study asphalt. The influence of different parameters was studied and analyzed. The manuscript lacks clarity and needs much improvement before further processing. This reviewer recommends minor editing and resubmits for re-review.

Comments:

  • The English writing of the manuscript needs improvement. Therefore, it could benefit greatly from professional editing to improve technical writing and English.
  • The authors should provide detail report on the existing studies. Literature section must be extended.
  • Please mention your study limits and suggest some future research topics
  • In References, the sources are written in different styles. It is necessary to bring in accordance with the requirements of the magazine for the design of References. If possible, indicate DOI.
  • Please use some innovative keywords.
  • Please mention your study limits in the abstract.
  • In the introduction, when formulating the problem and research objectives, it would not be superfluous to give the Ishikawa causal diagram with an analysis of the causes and factors of material degradation during freezing and thawing. This will strengthen the analytical content of the article and attract the attention of readers.
  • The Conclusions should reflect what the practical application of the results obtained in this study is. For example, what types of macadams and roads can be improved using the research results? In what climatic conditions should the recommendations of the authors be taken into account?
  • The authors should increase their discussion on previous related research and highlight how their study is providing a different approach or adding significantly to what has been done. The authors have to explain what is the new here in comparison with the previous studies. The novelty of the current work should be highlighted in the introduction. Please try to mention a problem that needs solving - in other words, the research question underlying your study clearer.
  • The title of the manuscript should be revised.
  • Some types of standards should be used to perform different experimental studies. Please provide details for the standards used in each study.
  • Section 4 should be discussed in detail.
  • The authors must redo the Abstract and bring it in compliance with the requirements of the Materials journal. The scientific problem is poorly described (Background). The scientific novelty is not indicated. I recommend shortening the Abstract to 200 words. Editors strongly encourage authors to use the following style of structured abstracts, but without headings: (1) Background: Place the question addressed in a broad context and highlight the purpose of the study; (2) Methods: Briefly describe the main methods or treatments applied; (3) Results: Summarize the article's main findings; and (4) Conclusions: Indicate the main conclusions or interpretations. The abstract should be an objective representation of the article
  • It is advisable to add a flowchart at the beginning of the paper. Then the article would become more visual and structured
  • Figure 10 is of poor quality. Replace, if possible, please
  • The conclusion should be an objective summary of the most important findings in response to the specific research question or hypothesis. A good conclusion states the principal topic, key arguments and counterpoint, and might suggest future research. It is important to understand the methodological robustness of your study design and report your findings accordingly. Please improve your conclusion section.
  • Please provide line numbers in the revised manuscript.

Reviewer 4 Report

Report on the manuscript

Title:  Evaluation of adhesion improvement measures between granite and
asphalt and pavement performance of asphalt mixture
Yali Yea, Yan Haoa, Chuanyi Zhuanga,*, Shiqi Shub and Fengli Lva

 Manuscript ID: Materials - 1545712

In the paper is evaluated the effects of anti-stripping measures such as silane coupling agent, rock asphalt modified asphalt, SBS modified asphalt, silane coupling agent and rock asphalt composite modified asphalt on the adhesion of granite aggregate and the performance of granite
asphalt mixture, on the physical properties. The paper is a description of some experimental results obtained by the authors. There is no calculation model, no estimates can be made on the results of new combinations, it is only a collection of experimental data. Generally speaking, the manuscript is well written, the material is judiciously divided and organized and correct from scientific point of view. Some changes are, however, necessary. For these reasons I can recommend the acceptance of this paper after some corrections.

Before that the Editor makes a decision, I suggest that the authors emphasize take into account the following corrections

  1. First of all, please respect the Template of the journal.
  2. The paper is written carelessly, a complete revision to bring it to a convenient form is necessary.
  3. The Abstract section is too long and with unnecessary information. Please present in the abstract what are you really do in the paper, what is the originality of the paper. 
  4. Please highlight how the work advances or increments the field from the present state of knowledge and provide a clear justification for your work.
  5. The section Conclusions will be point out the original results of the paper and can be extended to highlight the contributions. Please provide a clear justification for your work in this section, and indicate uses and extensions if appropriate.
  6. The conclusion section has to be rewritten doing an effort to remark the main findings rather than summarizing the article content.
  7. Please check the paper again for any possible misprints.
  8. The text needs to be checked and revised by a native speaker or a language expert. You may consider (at your own cost) the use of a possible professional copyediting service
  9. I think the authors need to emphasize more clearly the contribution of the manuscript from a scientific point of view.

If the author takes into account these observations the work can be published.

Round 2

Reviewer 2 Report

Authors have revised the paper based on the previous comments. My suggestion, the clarity of the abstract and paper formatting (e.g. citation format) should be improved

Reviewer 3 Report

The author addressed all the comments and the article can be accepted in its present form. 

Author Response

Thank you very much for your first pertinent suggestion, which has improved my paper a lot. Have a good day and all the best to your work!

Reviewer 4 Report

No comments

Author Response

(The authors gave the same response as above.)
